# Dialysis as a Novel Adjuvant Treatment for Malignant Cancers

**DOI:** 10.3390/cancers14205054

**Published:** 2022-10-15

**Authors:** Sture Hobro, Anders Nilsson, Jan Sternby, Carl Öberg, Kristian Pietras, Håkan Axelson, Ana Carneiro, Sara Kinhult, Anders Christensson, Jonas Fors, Steven Maciejewski, Jason Knox, Innas Forsal, Linda Källquist, Viktoria Roos

**Affiliations:** 1Baxter International Inc., 226 43 Lund, Sweden; 2Department of Nephrology, Clinical Sciences, Skane University Hospital, Lund University, 221 85 Lund, Sweden; 3Department of Laboratory Medicine, Division of Translational Cancer Research, Lund University Cancer Centre, Medicon Village, 223 81 Lund, Sweden; 4Skane University Comprehensive Cancer Center, Skane University Hospital, Lund University, 222 41 Lund, Sweden; 5Department of Oncology, Skane University Hospital, 221 85 Lund, Sweden; 6Department of Nephrology, Skane University Hospital, Lund University, 205 02 Malmö, Sweden; 7Baxter International Inc., Deerfield, MA 60015, USA; 8Baxter International Inc., Old Toongabbie 2146, Australia

**Keywords:** dialysis, HDAC, ketone bodies, immunotherapies, radiotherapies, chemotherapies, redox balance, cancer

## Abstract

**Simple Summary:**

There is a clear need for new cancer therapies as many cancers have a very short long-term survival rate. For most advanced cancers, therapy resistance limits the benefit of any single-agent chemotherapy, radiotherapy, or immunotherapy. Cancer cells show a greater dependence on glucose and glutamine as fuel than healthy cells do. In this article, we propose using 4- to 8-h dialysis treatments to change the blood composition, i.e., lowering glucose and glutamine levels, and elevating ketone levels—thereby disrupting major metabolic pathways important for cancer cell survival. The dialysis’ impact on cancer cells include not only metabolic effects, but also redox balance, immunological, and epigenetic effects. These pleiotropic effects could potentially enhance the effectiveness of traditional cancer treatments, such as radiotherapies, chemotherapies, and immunotherapies—resulting in improved outcomes and longer survival rates for cancer patients.

**Abstract:**

Cancer metabolism is characterized by an increased utilization of fermentable fuels, such as glucose and glutamine, which support cancer cell survival by increasing resistance to both oxidative stress and the inherent immune system in humans. Dialysis has the power to shift the patient from a state dependent on glucose and glutamine to a ketogenic condition (KC) combined with low glutamine levels—thereby forcing ATP production through the Krebs cycle. By the force of dialysis, the cancer cells will be deprived of their preferred fermentable fuels, disrupting major metabolic pathways important for the ability of the cancer cells to survive. Dialysis has the potential to reduce glucose levels below physiological levels, concurrently increase blood ketone body levels and reduce glutamine levels, which may further reinforce the impact of the KC. Importantly, ketones also induce epigenetic changes imposed by histone deacetylates (HDAC) activity (Class I and Class IIa) known to play an important role in cancer metabolism. Thus, dialysis could be an impactful and safe adjuvant treatment, sensitizing cancer cells to traditional cancer treatments (TCTs), potentially making these significantly more efficient.

## 1. Introduction

Reprogramming of energy metabolism is a hallmark of cancer cells [1], which is well known to support their rapid proliferation. Survival is an evident cancer cell need and four major metabolic changes related to survival have been identified: first, glycolysis is increased and has been shown to support the reduction of oxidative stress through multiple pathways linked to glycolysis; second, glutaminolysis has been shown to fuel other metabolic pathways also essential for the reduction of oxidative stress; third, both glycolysis and glutaminolysis fuel pathways that support the suppression of the host’s immune system in the tumor micro environment (TME), essential for cancer survival; and fourth, histone de-acetylases (HDAC) has been shown to be overexpressed in cancer cells and linked to cancer survival by improving the resilience of malignant cells to apoptosis and inducing immune suppression.

Ketone bodies serve as an excellent energy fuel for healthy cells and are primarily used for ATP production through the Krebs cycle in the mitochondria, utilizing OXPHOS and oxygen when other energy sources are limited. This stands in contrast to glucose and glutamine, which can be anaerobically used for numerous metabolites important for cancer survival and progression including energy production (ATP) [2,3,4,5]. Interestingly, ketone bodies have been shown to be inhibitors of class I and IIa HDACs and reduce several epigenetic alterations important for cancer survival [6,7,8].

Many cancers have a very short long-term survival rate, the 5-year survival rate is <10% for glioblastoma [9] and pancreatic cancer [10]. For most advanced cancers, therapy resistance limits the benefit of any single-agent chemotherapy, radiotherapy, or immunotherapy [11,12,13,14,15]. Therefore, some 5000 clinical trials are ongoing globally to probe the clinical benefit of new combination treatments [16]. The cancer cell resistance towards TCT has many causes [11,17,18]. An important one is the plasticity of tumor cells that may lead to the development of drug resistance, and therefore, the need for new cancer therapies targeting a multitude of mechanisms [16].

The novel concept described in this article suggests using dialysis, to reduce levels of both glucose and glutamine, as an adjuvant cancer treatment to increase efficiency and reduce side effects of traditional cancer treatments (TCTs). The goal with the suggested dialysis treatment is to timely shift the patient’s metabolic state, usually depending on glucose and glutamine, to another physiological state (ketogenic condition (KC) and low glutamine levels)—where all cells, including tumor cells, mainly must depend on ketone bodies for their cell energy metabolism. Such a dialysis treatment will induce a potentially harmful metabolic condition for cancer cells, mimicking the physiological KC obtained after fasting for many days or even weeks [19,20].

Dialysis is able to decrease the systemic levels of glucose and glutamine, two main nutrients, essential for cancer cell survival and proliferation, and at the same time support increased ketone bodies in the blood to secure metabolites for non-cancer cell ATP need through OXPHOS (Figure 1) [4,21,22,23,24].

To achieve a ketogenic condition with low glutamine levels, in conjunction with TCT, may boost the effectiveness of those treatments in similar ways as ketogenic diet and/or different types of glycolysis or glutaminolysis inhibitors are known to do [4,5,25].

Notably, a strict ketogenic diet would need to be maintained for days or even weeks and would naturally lead to a less pronounced metabolic shift (Figure 1) [19,20,26]. Additionally, several studies have shown that ketogenic diet has very poor cancer patient compliance and is hard to follow over time [19,27]. Dialysis, however, will have the power to create a significantly more prominent and well-defined metabolic shift compared with that of the ketogenic diet—and thus a more pronounced clinical effect (Figure 1) [28].

With the exception of lipids, all metabolic fuels in the human blood such as glucose, ketone bodies (KB), lactate, and amino acids (AAs), are small water-soluble substances that can be readily exchanged during dialysis. With dialysis, glucose [22] and glutamine [21] levels will be reduced, and simultaneous addition of KB (orally, intravenously, or by adding them to the dialysis fluid), could increase KB levels within a range of 2–8 mmol/L in blood [30] within one to two hours. Notably, ketoacidosis is a serious side effect of ketogenesis (not the presence of KB per se) that mainly occurs in type 1 diabetes. Ketoacidosis will not be induced when external KB is supplied [31]—thus avoiding this unwanted side effect.

## 2. Dialysis as a Cancer Treatment

Abnormal metabolism in cancer cells has been observed for close to a century and is today after almost 100 years recognized as a hallmark of cancer [1]. Cancer cells rewire their metabolism, and two common features of this altered metabolism are increased dependence on glucose and glutamine; exactly how and why cancer cells do this is not fully resolved. Nonetheless, this metabolic rewiring has multiple functions in cancer, including bolstering of cancer cell survival—resulting in an increased resistance to TCTs.

By changing the metabolite levels with dialysis as suggested, the cancer environment will change and consequently several anticipated anti-cancerogenic effects may occur (Figure 2). Thus, the changes imposed by dialysis may amplify the effectiveness of TCT. Increased KBs and reduced glucose and glutamine levels may increase the cancer cell’s dependency on mitochondrial ATP production and reduce the ability to handle oxidative stress by upregulating the pentose phosphate pathway (PPP), the one carbon metabolism (1CM), and glutaminolysis [4,5,23,24,32]. Reduced glucose and glutamine levels and increased ketone body metabolism lead to increased mitochondrial ATP production and have been shown to reduce oxidative stress in healthy cells [32,33,34] and increase oxidative stress in cancer [2,34,35,36]. By reducing lactate and increasing pH in the TME, the immunologic response to cancer cells may increase [37,38]. Furthermore, KBs have, at physiological levels (typical for fasting), been shown to inhibit HDAC class I and class IIa, and many cancers show an increased expression of these HDACs [6,8,39,40,41].

Consequently, to force KC by dialysis during a four-to-eight-hour treatment will induce a multitude of changes potentially harmful for cancer—and if done in conjunction with TCTs, potentially augment their efficacy.

## 3. Dialysis Treatment and Its Potential Use as an Adjuvant Cancer Therapy

### 3.1. How Kidney Dialysis Is Used Today

Chronic kidney disease affects one in ten people globally, of which about one percent require kidney replacement therapy (KRT) in the form of dialysis or kidney transplantation. Institutional hemodialysis (Figure 3A) remains the most common form of KRT and is typically given as four-hour treatments in-hospital thrice weekly. Blood is supplied to an extracorporeal circuit from either a central dialysis catheter or an arteriovenous fistula and is pumped at a rate of 200–400 mL/min to a dialyzer composed of a blood compartment and a dialysis fluid compartment separated by a semi-permeable membrane. The fresh dialysis fluid is mixed in a dialysis machine with water and chemicals in dry or liquid form to concentrations desired for the patient’s blood. In many parts of the world, glucose is added to the dialysis fluid to avoid hypoglycemia, especially in diabetic patients [42,43,44,45]. The dialysis fluid flow is typically twice the blood flow. Metabolic waste solutes are transferred from the blood to the dialysis fluid across the membrane mainly via diffusion.

In the case of CancerDialysis, the blood is suitably accessed by a double lumen central venous catheter (CVC) inserted in a large vein, preferably vena cava via the internal jugular vein. The lumen size is somewhat larger than a standard CVC to allow required flows. If the catheter needs to be maintained for more than a week, it is preferably tunneled to minimize the risk of infections (Figure 3B).

Hemodialysis is very effective for small water-soluble substances such as electrolytes, urea, and creatinine. Small water-soluble molecules are rapidly exchanged over the semipermeable dialysis membrane, and their levels in the blood leaving the dialyzer are similar (85–95%) to those in the dialysis fluid [48]. Hemodialysis treatments typically have minor side effects, most common during treatment are hypotension, muscle cramps and headache [47].

In order to achieve a starvation-like metabolic state with dialysis, a carbohydrate-restricted diet would be required 12 to 18 h before a glucose free dialysis treatment starts [49,50]. However, other sources of energy are encouraged, and patients could even receive appropriate parenteral nutrition to compensate for glucose and amino acid loss during the carbohydrate-restricted diet and the succeeding dialysis treatment. The goal of dialysis therapy is to mimic the metabolic condition after that of prolonged fasting for weeks or following long-endurance exercise (many hours).

It is well-known that glucose levels in the blood are reduced when dialysis fluid without glucose is used (still the practice in large parts of the world). During a standard dialysis session of four hours without glucose in the dialysis fluid, the glucose level can be reduced towards 3 mmol/L in fasting patients [22]. Any small water-soluble molecules present in the blood plasma, such as glucose and AAs (including glutamine), are markedly reduced during a normal dialysis session [21]. In fact, the idea to use dialysis to treat cancer is not new. For example, Mathews et al. [51] suggested the use of dialysis to remove glucose and glutamine to a point where almost no glucose (below 0.45 mmol/L or 0 glucose) is left in the blood. It is hypothesized that cancer cells will die if they are suddenly deprived of glucose, due to their strong dependency on glucose metabolism [24,51]. However, such low glucose levels constitute not only a risk for the cancer but certainly also for the patient. Thus, in the idea described by Mathews et al., the focus is on supervision and monitoring the patient to ensure safety during low glucose plasma levels.

During hemodialysis the plasma water in the patient will gradually change its composition toward that of the dialysis fluid. A four-hour dialysis session for a dialysis patient is typically dimensioned to clear 75% of dialyzable waste products from the whole body. Organs that are well penetrated with blood are cleared faster. Assuming that vital organs receive 70% of the blood pumped by the heart, and that they contain 20% of the total distribution volume of a dialyzable substance [52], they will be 75% cleared after circa 70 min. Thus, the time to get an effective anti-cancer blood environment by dialysis depends on where the cancer is situated and how well the tumor tissue is perfused.

Using positron emission tomography (PET), glucose- or glutamine-consuming cancers may be identified. Typically, the radioactive tracer that is incorporated (often glucose) is infused for 60 min before performing the PET scanning, indicating a 60-min turnover rate for glucose [53]. The suggested time on dialysis in order to ‘treat’ cancer is 4–8 h; the reduction towards 3 mmol/L in blood glucose level will occur during the first 30 min of dialysis, requiring that the glycogen stores in the liver are depleted when the dialysis session starts.

Simulation of dialysis on a system level (Figure 4) shows that blood glucose levels drop rapidly after dialysis is started if glycogen levels in the liver are depleted before treatment.

Consequently, CancerDialysis can promptly switch the metabolic conditions for the cells.

### 3.2. When and How to Apply CancerDialysis

CancerDialysis may be used before, during, or after a TCT. In the case of chemotherapy, the loss of the active agent must be compensated for when CancerDialysis is performed—during or directly after such a treatment. The loss of drugs during dialysis treatment is well known in both chronic dialysis and acute dialysis [56,57]. Possibly, a cytotoxic agent may be added to the dialysis fluid as suggested for antibiotics during CRRT in sepsis—either to compensate for the loss of the drug this way or to fully administer the cytotoxic agent in the dialysis fluid, thereby adding the benefit of controlling the levels of the agent in the blood throughout the CancerDialysis treatment [56,58]. Nevertheless, the best opportunity for applying CancerDialysis could be closely in time after the TCT.

Zhong et al. showed that “Radiation induces an increase in tumor glucose demand approximately 30 h following therapy during reoxygenation” [59]. Therefore, to do CancerDialysis after radiotherapy may be the best option with the rationale that after radiotherapy cancer cells will be deprived of their propensity to use glucose and glutamine to reduce oxidative stress when most needed, see Section 3.3.2. Possible schemes could be to do radiation therapy for five consecutive days of the week, using the sixth day for a CancerDialysis treatment or using a 3-day rotating schedule with two days of chemo/radiotherapy and CancerDialysis treatment every third day.

For immunotherapies, our simulation of the TME during dialysis indicates a more “immune-friendly” environment during dialysis. For immunotherapies, the suggestion may be to perform CancerDialysis a number of times some days after the immunotherapy treatment is performed, supporting the immune cells to be active in the TME and starting the process to induce cancer cell apoptosis, see Section 3.3.3.

Consequently, the optimal way to utilize CancerDialysis remains to be explored; this must be experimentally determined by in vivo tests and in human clinical trials.

### 3.3. Effects of CancerDialysis

During normal feeding conditions, glucose plays a vital role in producing the ATP needed to sustain human cell energy metabolism. However, during periods of fasting or low carbohydrate diet, humans can switch from their dependency on glucose to rely on ketones for their energy metabolism. Similar to hybrid cars, which can switch from dependence on liquid fuels to electricity. However, the switch from dependence of glucose to ketone bodies requires extensive changes in cell metabolism, a talent that human cells have developed over millennia. Below follows a more detailed summary on how this metabolic switch imposed by CancerDialysis (Figure 5) may support TCTs.

#### 3.3.1. Metabolic Effects

A hallmark of cancer cells is a reprogrammed energy metabolism where cancer cells ferment and overutilize glucose and glutamine to produce energy and building blocks, and thereby reduce the dependence on the mitochondria for energy production [4,23,60]. Thereby, cancer cells derive their ATP from overconsumption of anaerobic glycolysis. Conversely, healthy cells use the more efficient aerobic pathways in their mitochondria to produce ATP [4,28,61].

Both glucose and glutamine fuel multi-branched metabolic pathways (Figure 6) are of large importance for cancer cells, providing large metabolic flexibility, which may contribute to treatment resistance and evasion of apoptosis in cancer [62,63,64,65].

Glucose via glycolysis fuels both the pentose phosphate pathway (PPP) and the one carbon metabolism (1CM)—both overutilized in most cancers [66,67,68]. The high dependency on glycolysis in cancer cells leads to increased production of lactate and hydrogen ions, which are exported to the TME, resulting in increased lactate levels and pH reduction in the TME [37,69,70].

Glutamine, which fuels several fermentable pathways, is also important in malignant cells and can lead to the production of lactate, which in turn is exported to the TME and generates NADPH through malic enzyme; also, the isocitrate dehydrogenase enzyme can contribute to lactate production in cancer cells, increasingly important for cancer cells during glucose deprivation [23,71,72,73].

If glutamine and glucose are reduced concomitantly, many complex and interlinked pathways important for cancer cells’ survival will be affected. One of them is the functionality of glutathione, which is crucial for the survival of cancer cells and resistance to traditional cancer therapies [11,74,75]. Some studies indicate the efficacy of the combination of reduction of both glucose (with KD) and glutamine (glutaminolysis inhibitors) during TCTs [24,60].

A remarkable ability of the body to adapt to long-term starvation is the utilization of stored fat, critical for human survival during periods of prolonged starvation. The most striking alteration during this adaptation is the ketone body (KB) production and utilization (Figure 7). Produced by the liver, KBs can be used by the majority of all human cells as a fuel. KBs also exert signaling effects and a KC inhibits glycolysis and gluconeogenesis in addition to merely glucose reduction [82,83,84]. Furthermore, glutamine uptake and utilization may be reduced when KC is implemented [83,84].

Some cancer cells lack the enzymes needed to metabolize ketones and others have altered and impaired mitochondria function; hence, if glucose and glutamine levels are reduced, energy levels may fall below critical levels and induce apoptosis [2,28,35,39,87,88]. Additionally, it has been shown that adding ketone bodies in vitro reduces utilization of glucose and glutamine uptake directly in malignant cells [83].

Healthy cells can effectively use ketones as a metabolic energy fuel when they utilize oxygen and the mitochondria for ATP production [28,89]. This ability to use oxygen and the mitochondria for ATP production seems to be limited in cancer cells [28,39,89,90].

Consequently, during a ketogenic and low glutamine condition, induced by dialysis, cells must depend on the Krebs cycle for their energy metabolism. Thus, the metabolic flexibility crucial for cancer survival, characterized by usage of the glycolysis and glutaminolysis pathways, will be significantly reduced following CancerDialysis.

#### 3.3.2. Redox Balance Effects

Elevated levels of reactive oxygen species (ROS) are characteristic of cancer [2,36,74,91]. To prevent oxidative damage, cancer cells adapt metabolically to maintain the balance between reduction and oxidation (redox balance) by increasing their production of reducing equivalents to handle the increased ROS produced [23,74,91]. This balance plays an important role in the regulation of cancer cell survival [23,91]. In general, moderate levels of ROS may promote cell proliferation and survival, whereas a more pronounced increase of ROS can induce cell death [23,32,74,91]. The increased generation of ROS in cancer is a treatment opportunity and is utilized by several TCTs such as radiotherapy and most chemotherapies; they indeed induce ROS production in the cancer cells [23,28,34,36,76].

Counterintuitively, more oxidative stress is generated during hypoxic condition than during normoxia. As a result, hypoxic tumors rely heavily on antioxidant systems to sustain ROS homeostasis [18,92]. Altered energy metabolism, also in the presence of oxygen (the Warburg effect), is one of the hallmarks of cancer, in which the metabolism is shifted from oxidative metabolism towards anaerobic glycolysis. This metabolic alteration provides cancer cells with abundant substrates for increased reductive power to uphold the redox balance [79,93].

The most important reduction molecule in cancer is NADPH, which is crucial for the redox balance in cancer cells; it is the essential electron donor providing reducing power to several pathways controlling the redox balance in cancer. Specifically, NADPH reduces the central player of redox cell balance, glutathione, which has an indispensable role in maintaining redox hemostasis in cancer cells [23,74,75,79,93]. NADPH synthesis is increased in cancer cells and NADPH can be produced through many different pathways, whereof the most important emanates from glycolysis, and the pentose phosphate pathway, which is fueled by G6P, a metabolite in glycolysis (Figure 6) [32,79]. However, during glucose deprivation, several glutamine-dependent pathways become increasingly important [23,60,80]. For example, malic and isocitrate dehydrogenase enzymes can produce NADPH in the absence of glucose, and are predominantly driven by glutamine, and becomes increasingly important for NADPH generation during glucose deprivation [23,77,79].

Thus, if the access to glucose and glutamine is significantly reduced, the cancer cell will have limited ability to produce reductive power in the form of NADPH to handle increased oxidative stress [23,79,80,83] (Figure 6).

A KC will reduce glucose availability, and if the cancer cells have the ability to utilize KB for ATP production this will further diminish glycolysis—and when such cancer cells are forced to use ketone bodies to produce energy through their often few, malformed, and dysfunctional mitochondria, ROS production is likely to increase [2,34,94]. Reduced glycolysis will reduce the availability of substrate in the glycolysis pathway and reduce the cancer cells’ ability to produce NADPH through PPP. Thus, by creating a KC and simultaneously reducing glutamine levels through dialysis, the possibility for a cancer cell to maintain its redox balance is severely impacted in two different ways: (1) the increased utilization of the mitochondria might increase ROS levels, and (2) the ability to handle ROS through increased glycolysis and glutaminolysis, and increased NADPH production, will be decreased [2,34] (Figure 6).

In healthy cells, the shift to ketone bodies as a fuel for mitochondrial energy production is linked to reduced oxidative stress [28,33,82]. This effect is mediated by different mechanisms not involving NADPH [6]. Firstly, the need to utilize NAD^+^ (another substance that can reduce ROS) to produce acetyl-CoA when ketone bodies are used to feed the Krebs Cycle is reduced compared with that when using glycolysis [33,95,96,97,98]. Secondly, healthy cells can reduce the ROS produced in the mitochondria from ATP production, by increased uncoupling in the electron chain transfer [64,99]. Thirdly, ketone bodies have a pivotal role as a signaling mediator and modulates oxidative stress in healthy cells [5,33,59]. All those functionalities tend to be more limited in cancer cells, and consequently increased levels of ketone bodies in cancer cells seem to induce oxidative stress in those [2,34,76,98,100,101].

Several studies suggest that reduced glycolysis and increased dependency on ketolysis lead to reduced side effects in healthy cells upon treatment with radio- and chemotherapy [2,28,34,35,89].

Consequently, these differences between how healthy cells and cancer cells react to a KC act as a metabolic wedge during CancerDialysis, where ROS is reduced in healthy cells while it is increased in cancer cells during KC. Therefore, we suggest that reducing glucose and glutamine levels with dialysis and simultaneously increasing ketone bodies in the blood may work as an adjuvant treatment to radio- and chemotherapies [2,28,35,36,60,89]. Conceivably, this strategy would augment the effect of TCT by increasing ROS in cancer cells while simultaneously impairing the ability of cancer cells to handle the ROS, and concurrently reducing the impact of TCT on healthy cells [28,35,36].

#### 3.3.3. Immunological Effects

As described earlier, an altered metabolism is a new hallmark of cancer. A second more recently added hallmark is the ability of cancer to avoid the immune response [1]. The TME is harsh for the infiltrating immune cells, and to be effective they need not only to survive, but also function well enough to kill the cancer cells. This environment is to a large extent shaped and influenced by the tumor metabolism, acting as a defense against the infiltrating leukocytes (Figure 8) [15,37,38,69,102,103].

Immunotherapy is now an integral and important treatment modality for the treatment of many types of cancers. Immune cells utilize both glucose and glutamine during normal conditions. However, it has been shown that immune cells are metabolically flexible and can be effective also during a KC [109] and at low glutamine levels [102]. Interestingly, Mukherjee et al. demonstrated total remission of glioblastoma in a mouse model by combining KD with a glutamine inhibitor, constituting compelling support for a dual strategy to target both these fuels [60]. Thus, this suggests that high lactate levels and low pH in the TME are a larger barrier to the proper function of the immune system in cancer than low glucose and glutamine levels [70,102,109,110].

A result of the excessive use of glycolysis and glutaminolysis in cancers is the increased production of lactate and hydrogen ions that are exported out from the cancer cells [38,69,104,111]. This creates very high lactate and low pH environment surrounding the cancer cells, resulting in important immunosuppressive effects by incapacitating leucocytes, such as natural killer cells (NK-cells) and CD8 T cells that are known to be inactive within the harsh conditions of the TME [38,104,106,107,111].

KD is known to reduce circulating blood glucose levels [112,113,114]. Increasing ketone bodies in vitro independently reduces glycolysis beyond just the reduction of glucose [83,87]. Therefore, KC also contributes to reduced glycolysis in cancer cells and in turn reduced lactate levels and increased pH in the TME [83,87]—which in turn may promote immune responsiveness [5,70,83,104,105]. This suggests that using CancerDialysis to enforce a KC a few times per week may be a better way to induce a response from immunotherapies than using KD [104]. Besides a direct ameliorating effect towards immunotherapies, radio- and chemotherapies may also benefit from a more active immune system induced by a KC [38,69].

One study performed by Ferrere et al. showed that intermittent fasting increased the effect of immune checkpoint blockers and also indicated that intermittent administration of KBs may have similar effects as that of a continuous KD [109]. This suggests that using CancerDialysis to enforce a KC a few times per week may be a better way to induce response from immunotherapies than using KD.

Leone et al. showed that targeting glutamine metabolism in cancer cells leads to wide-ranging metabolic inhibition, disruption of NADP(H) homeostasis, and stymied growth. Conversely, targeting glutamine metabolism in T cells leads to adaptive metabolic reprogramming with enhanced survival, proliferation, and effector function [102].

Another opportunity created with dialysis is to optimize the buffer capacity in the cancer patient by increasing systemic bicarbonate levels. Increased systemic buffer capacity (i.e., increased bicarbonate) also increases local buffer capacity and increases pH in the TME—and is shown to enhance immunotherapy response as well as other TCT responses [37,108,115].

In chronic dialysis patients, the loss of kidney function and its capacity to regulate bicarbonate is lost and therefore bicarbonate must be replenished at each dialysis treatment. Typically, the levels in the blood are increased to 28–30 mmol/L after a hemodialysis treatment (physiological levels: 22–26 mmol/L). However, for some patients (typically younger and more active) it needs to be increased to levels up to 35 mmol/L [116]. Ingestion or infusion of bicarbonate may lead to life-threatening adverse side effects such as hypokalemia, and the risk-to-benefit ratio for such an action may be too high [37,117]. During dialysis, the ion composition in the blood is controlled and severe side effects from ion imbalances from the raised bicarbonate are actively avoided [118].

In conclusion, dialysis as suggested may work as an adjuvant treatment to immunotherapies by reducing the immunosuppressive functions of cancer cells, while simultaneously taking advantage of the metabolic and redox balance effects as described above.

#### 3.3.4. Epigenetic Effects

Yet, another recently proposed emerging hallmark of cancer is epigenetic reprogramming. Epigenetic changes influence cancer cells in numerous ways and have attracted growing interest during the last decade [1,40,119]. It can be anticipated that epigenetic reprogramming is involved in enabling several hallmarks of cancers [1]. Epigenetic alterations may exist in all cancers and are most known to be induced by an increase in histone deacetylases (HDAC) [40,120]. Histone deacetylases inhibitors (HDACi) are a new type of drug, reducing the epigenetic changes from increased histone deacetylases HDAC [41,119,121,122].

Several epigenetic drugs have been approved to treat different cancers [40,119,123]. Furthermore, the use of those drugs in combination with chemotherapy or immunotherapy has shown compelling outcomes, including augmentation of anti-tumoral effects, overcoming drug resistance, and activation of host immune response [119,122,124].

Besides ketone bodies functioning as a fuel for mitochondrial energy production, they also show an important function as pan-HDACi and modulate the expression of numerous genes [36,87,125,126]. Ketone bodies inhibit several different types of HDACs classes I and IIa (HDAC 1-9, except HDAC 6) (Figure 9), at levels achieved during ketogenic diet and/or fasting [6,39,70,95,125,126,127,128].

Mikamin et al. found that beta-hydroxybutyrate (the most abundant KB) significantly enhanced cisplatin-induced apoptosis in liver hepatocellular (HepG2) cells, via HDAC silencing through enhanced cisplatin-induced apoptosis via inhibition/surviving axis [8]. Lee et al. found that inhibition of HDACs 3 and 6 had similar effects in breast cancer cells [130]. The reduction of HDAC overexpression may therefore further augment radio- and chemotherapy treatments intended to increase ROS in cancer [8,129]—or immunotherapies to induce immunological responses in cancers [121,122,124].

Taken together, CancerDialysis as suggested here may further augment immunotherapies, as well as radio- and chemotherapies, by reducing HDAC overactivity in cancer, promoting key processes including cell-cycle arrest, apoptosis induction, and upregulation of tumor suppressors [40,41].

## 4. Future Endeavors

To evaluate if hemodialysis as an augmenting treatment may improve survival for malignant cancers with poor prognosis.

To verify the CancerDialysis hypotesis, in vivo studies in the C6/Wistar rat gliom model could be a possible way forward. The Wistar rat is immuno competent and the possible immuno supporting effects from CancerDialysis may therefore also be demonstrated in Wistar rats. Typically, such studies could give answers on when, how long, and how often CancerDialysis should be applied together with TCTs [19,27,34,36,76,83,88,100,101,126].

Importantly, the potential of the CancerDialysis concept would need to be verified in human clinical trials, where it is suggested to start studies on patients with cancer types where the ketogenic diet has shown compelling results in enhancing TCTs. This includes cancer types with poor survival rates and where no or few effective treatment options exists, such as glioblastoma [9,14,28,34,36,131,132], pancreatic cancer [10,19,34,83,88,98], and other more or less treatment resistant cancers [19,34,36,76].

A pleiotropic treatment option such as CancerDialysis may make such cancer types more vulnerable towards TCTs, by reducing important cancer characteristics such as increased antioxidative capacity, increased immune suppression, and increased expression of various oncogens of HDAC.

## 5. Conclusions

It is conceivable that one way to combat therapy resistance in cancer could be targeting the “engine” of the tumor survival and growth, i.e., targeting its driving metabolism. Independently of the oncogenic drivers, metabolic alteration is the means by which cancer cells bolster their survival. Thwarting the possibility to utilize those metabolic alterations by inducing a KC may therefore reduce therapy resistance and make TCTs significantly more effective. Utilizing dialysis to induce a KC and reduce glutamine levels in blood, making glucose- and glutamine-dependent cancer cells more vulnerable to TCTs, is an underexplored treatment option and may give hope to many cancer patients, especially those with no or little hope for long-term survival (Figure 2).

Dialysis treatment is performed worldwide millions of times per day; it is a safe and effective treatment with minor side effects—also after years of chronic dialysis three to four times per week. The ketogenic diet has been shown to be an effective treatment for epilepsy in children; promising results are also achieved in a variety of cancers when used in animal models. However, the few human studies performed to investigate the effect of a ketogenic diet on cancer patients show low diet compliance. Combining TCT with a dialysis treatment to affect multiple hallmarks of cancer constitutes an attractive approach to explore, and it has the power to circumvent the compliance problems that ketogenic diets experience. Additionally, dialysis offers the advantage of regulating the levels of the metabolites in a controlled way.

Today, only 15–20% of immunotherapy treatments are effective; combining these with dialysis may increase efficiency and improve the outcome. Furthermore, dialysis may make chemo- and radiotherapies more effective and simultaneously reduce the side effects on healthy cells [36].

The novel approach we present here, to use dialysis as described, can open new avenues for targeting many inherent vulnerabilities of cancer cells, including metabolism, ROS control, immune suppression, and epigenetic reprogramming. The metabolic condition induced by dialysis will be a multi-combinatoric treatment and potentially with few side effects, principally utilizing and mimicking an alternative metabolic state that our ancestors have used for millenniums to support their survival during periods of food shortages.

## Figures and Tables

**Figure 1 cancers-14-05054-f001:**
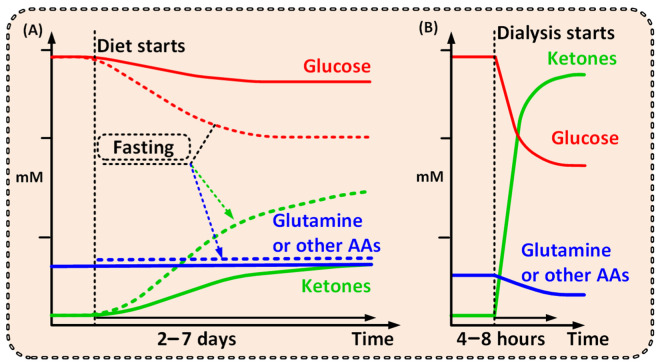
Graphs outlining the possible differences between the metabolite levels achieved over time from (**A**) a ketogenic diet [26,28] or fast [20,29], and from (**B**) dialysis. A comparably short dialysis treatment would shift the patient to a ketogenic metabolic condition, where high ketone and low glutamine levels will be unfavorable for cancer cells, but not healthy cells. A short dialysis treatment shifts the patient from one metabolic condition to another, unfavorable for the cancer but not for the patient as a whole.

**Figure 2 cancers-14-05054-f002:**
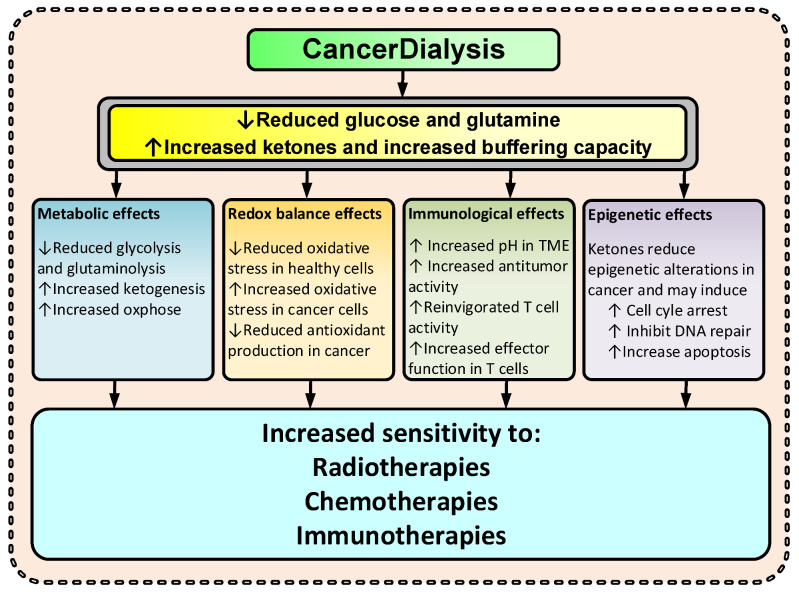
Suggested effects of CancerDialysis on the efficacy of TCT. (↑) indicates an increase and (↓) indicates a decrease.

**Figure 3 cancers-14-05054-f003:**
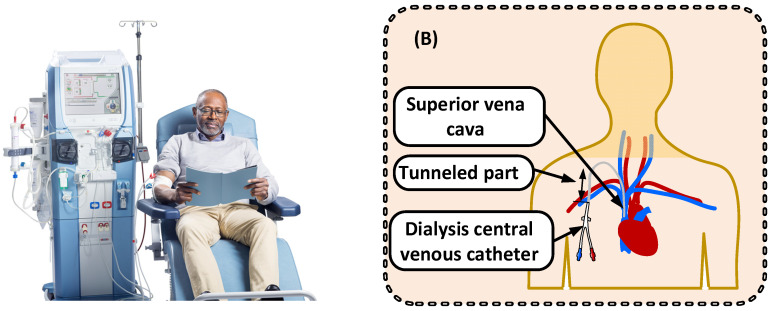
(**A**) Hemodialysis is the choice of renal replacement therapy, typically performed with a blood access in the arm (a fistula), for most patients that needs therapy for end stage renal disease, and in 2010 more than two million patients received dialysis treatment [46]. The treatment is generally well-tolerated with minor side effects [47]. (**B**) A tunneled central dialysis catheter is also used for chronic patients but also for temporary access to the blood. It allows for long-term access to the vein and is bidirectional, allowing flows in and out at the same time.

**Figure 4 cancers-14-05054-f004:**
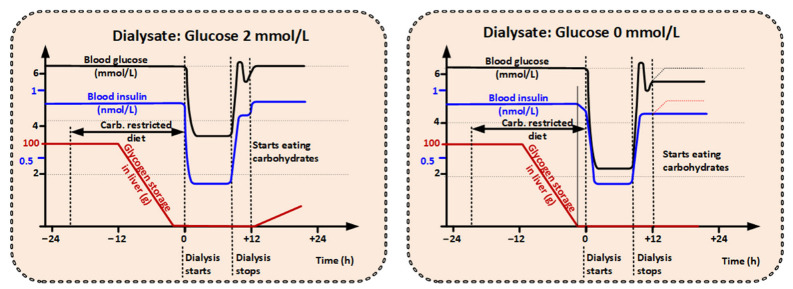
The dialysis simulation is utilizing a standard dialysis setting of 300 mL/min in blood flow and a 500 mL/min in dialysis fluid flow; the physiological model we have used for the simulation has been presented by Todd et al. [53,54]. When diet starts, the available glycogen level is set to 100 g in the liver [49,50,55]. When dialysis starts, the blood glucose level will fall towards 3 mmol/L if a dialysis fluid glucose concentration of 2 mmol/L is used; if the dialysis fluid glucose concentration is 0 mmol/L the simulation indicates a fall toward 2 mmol/L of glucose in the blood.

**Figure 5 cancers-14-05054-f005:**
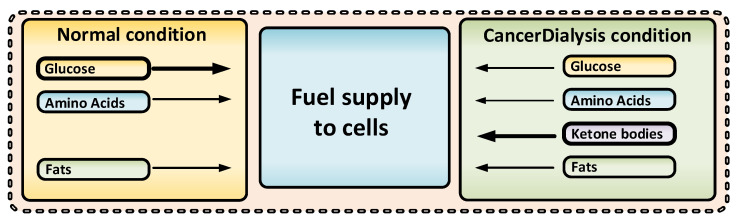
During CancerDialysis healthy cells as well as cancer cells (except hepatic and red blood cells) must shift from their dependency on glucose and amino acids towards ketones. This shift from a normal physiological state to a ketogenic condition, is a shift that healthy cells are programmed to do timely during food shortages, an ability that cancer cells, which are more addicted to fermentable fuels as glucose and glutamine, may be more vulnerable to. To induce KC and low glutamine levels with dialysis may therefore work as a multi-combinatorial treatment, potentially with few side effects when healthy cells timely can adapt to KC.

**Figure 6 cancers-14-05054-f006:**
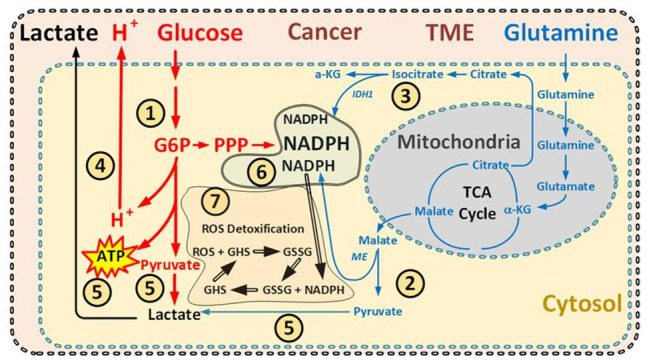
Schematic view over most important NADPH-related pathways in the cytosol and the major pathway for lactate and hydrogen ion production. NADPH plays a vital role in the reduction of the increased oxidative stress in cancer [23,76,77,78]. NADPH is known to be produced through several fermentable metabolic pathways, where most depend on glucose or glutamine [78,79]. ➀ The most important pathway for NADPH production is the pentose phosphate pathway (PPP) that emanates from glycolysis and depends on glucose for its function [78,79,80]. ➁ The second most important pathway for NAPDH production is conversion from malate to pyruvate, fueled by glutamine and increasingly important during glucose deprivation [62,79,80,81]. ➂ In some cancers there is another pathway for NADPH production, mainly glutamine fueled, utilizing isocitrate dehydrogenase [77,79,80]. ➃ Hydrogen ions are produced as a product of the glycolysis pathway (red) and are exported out from the cytosol, thereby contributing to the low pH in the TME [37,69]. ➄ Lactate can be produced either both from glucose (red) or from glutamine (blue) and is exported out from the cytosol (black) and contributes to high lactate levels in TME [37,69]. ➅ Schematically describing the NADPH pool in the cytosol and how it is constantly refilled from pathways mainly dependent on glucose and glutamine [11,79,81]. ➆ The reduction power from NADPH is used to detoxify the cells from oxidative stress. Glutathione (GSH) is one of the most important scavengers of reactive oxygen species (ROS) in cancer cells and when oxidized by ROS to glutathione disulfide (GSSG) it is reduced back to GSG by NADPH [11,74].

**Figure 7 cancers-14-05054-f007:**
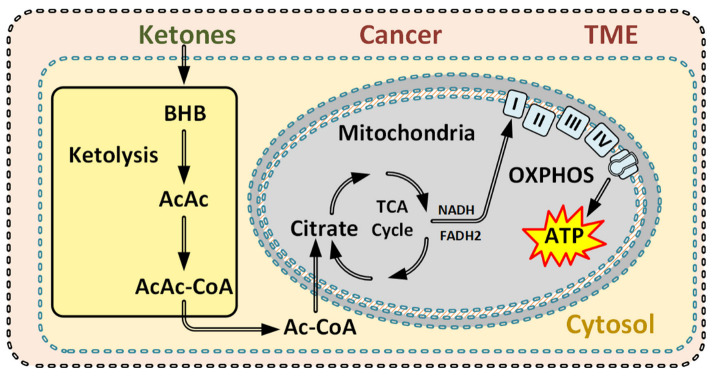
When ketone levels increase in the blood, the gluconeogenesis in the liver is reduced [85,86]. Ketolysis can occur in almost all healthy cells [39,82]. Ketones enter Krebs cycle through acetyl-CoA requiring oxaloacetate to enter [39,82]. Comparably, ketolysis is a short and straight pathway, and do not contribute to the production of anti-oxidative power or metabolites in cancer cells as glycolysis or glutaminolysis do (compare Figure 6).

**Figure 8 cancers-14-05054-f008:**
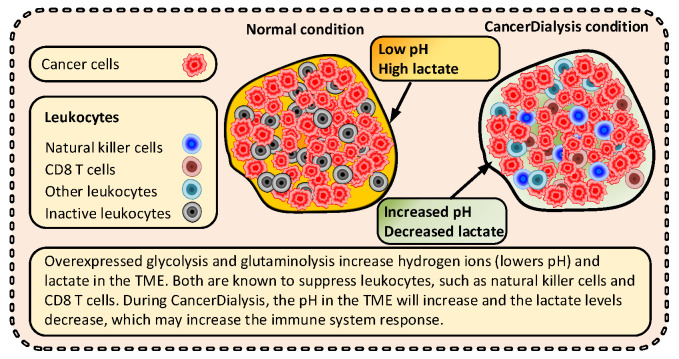
It is well known that the TME is a harsh environment for the infiltrating leukocytes. CD8 T cells and natural killer cells are important immune cells for targeting cancer cells and are known to be inactive at low pH and high lactate levels [15,37,70,104,105,106,107,108].

**Figure 9 cancers-14-05054-f009:**
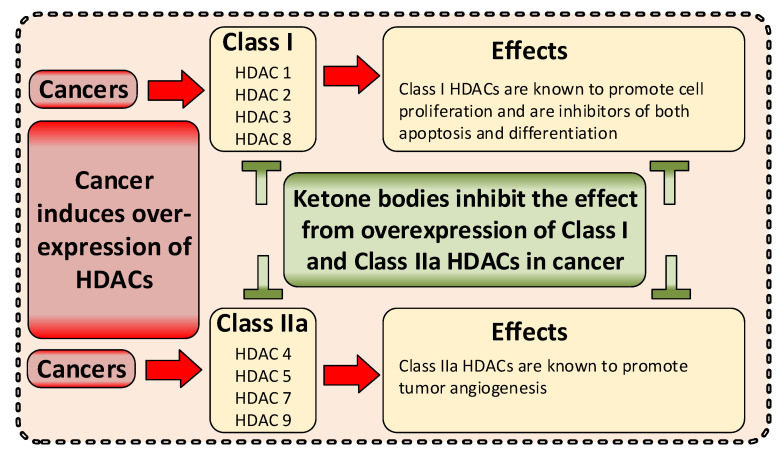
Elevated HDACs are linked to oncogenesis and HDACs have pleiotropic effects and promote many oncogenes. Reducing HDACs through HDAC inhibitors can improve therapeutic efficacy of TCTs. For example, the strong suppression of apoptosis seen in many cancer cells will be reduced through the inhibition effects that KBs are shown to induce at physiological levels [6,40,120,121,128,129].

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
