# Peer review of "Dialysis as a Novel Adjuvant Treatment for Malignant Cancers"

_cancers, 2022, doi:10.3390/cancers14205054_

Round 1

Reviewer 1 Report

The paper is well organized and describes the cancer-dialysis progress. 

Comments:

1. More figures are needed for section 3.3. The figures can be reproduced from the references. Figure 3 is not very informative, and it can be merged with figure 2.

2. Please check all section to make sure there will be a overall summary by  the end. It will make the whole structure clearer.

3. Please discuss more on the prognosis of cancer treatments. 

Author Response

Dear Sir/Madam, 

A sincere thank you for the very constructive comments you provided, and for rating our manuscript as interesting for future publication. We have now reviewed and acted on all comments provided and feel that the updated manuscript is significantly improved with added information according to your input. 

Specifically: 

1. More figures are needed for section 3.3. The figures can be reproduced from the references. Figure 3 is not very informative, and it can be merged with figure 2. Answer: Thanks for pointing this out and thereby enhancing the understanding of our work; we have now introduced new figures into section 3.3, added Figures 5, 8 and 9. We also decided to update Figure 3; we realized from your comment that the purpose of this figure was not clear and this has now been taken care of by clarifying the text in Section 3 emphasizing that hemodialysis is a relatively gentle treatment; moreover Figure 3 has been updated with more details and formatted caption. 

2. Please check all section to make sure there will be a overall summary by  the end. It will make the whole structure clearer. Answer: Again, we truly appreciate this comment that improves the clarity of our work and we have carefully gone through the manuscript and added summaries at the end of each section throughout the text. 

3. Please discuss more on the prognosis of cancer treatments. Answer: We agree and believe that this added information will put our work in perspective and highlight the importance of finding effective cancer treatments. We have now added a paragraph in Section 1/Introduction and also text regarding how to verify cancer dialysis (Section 4) and which cancer types to target. 

Yours sincerely, 
/Viktoria Roos 

Reviewer 2 Report

The scientific method requires that the hypotheses must be verified. This work poses interesting hypotheses but none of which is verified

Author Response

Dear Sir/Madam,

We thank you for your input and comments on our manuscript. We fully agree that our proposed concept is currently a hypothesis that needs to be further developed and verified in future studies to truly show its potential. We have made this more clear by providing a discussion on this topic adding a new Section 4 to our manuscript with the title ‘Future endeavors’. We thank you for finding our hypothesis interesting. 

Yours sincerely,
The Authors

Reviewer 3 Report

The paper reviews the dialysis treatments to treat malignant cancers. It describes several dialysis using different glucose concentration and studies the metabolite intermediate.

I considered that it is a well-written paper and can be very interesting

Author Response

Dear Sir/Madam,

We thank you for your input and for rating our manuscript as providing significant contribution to the scientific field  – and for finding it very interesting.

Yours sincerely,
The authors 

Round 2

Reviewer 2 Report

There are some typos and some citations are invalid and need to be corrected. I thank the authors for having added the final comment but I still think that the editorial position is more of a perspective than that of a review.

Author Response

Dear Sir/Madam,
We thank you again for your constructive feedback. We have thoroughly gone through the manuscript yet again and corrected all minor remaining spelling and grammar mistakes - the ones visible and also others, including corrections in the figures. We have also taken great care to update and correct the references. 
We respect your standpoint about this being a work that relates to a theory in need of validation; however, our work, which is of review character, is based on numerous well-cited and highly ranked scientific articles, which we have collected, studied well, synthetized, and which we base our novel concept on – and which we have performed mathematical physiological modeling on to verify the concept. We truly believe that our new innovative approach is worthy attention in the scientific community and deserves to be tested for the benefit of many of our patients world-wide. 

Yours sincerely, 
/The authors